# Climate change in a conceptual atmosphere–phytoplankton model

György Károly[1], Rudolf Dániel Prokaj[2], István Scheuring[3], and Tamás Tél[4]

[1]Budapest University of Technology and Economics, Institute of Nuclear Techniques, Műegyetem rkp. 3., 1111 Budapest, Hungary
[2]Budapest University of Technology and Economics, Department of Stochastics, Műegyetem rkp. 3., 1111 Budapest, Hungary
[3]MTA-ELTE Theoretical Biology and Evolutionray Ecology Research Group, Department of Plant Taxonomy, Ecology and Theoretical Biology, Eötvös University, Budapest, Hungary and MTA Centre for Ecological Research, Evolutionary Systems Research Group, Tihany, Hungary
[4]Department of Theoretical Physics, Eötvös University, Budapest, Hungary, and MTA-ELTE Theoretical Physics Research Group, Budapest, Hungary

**Correspondence:** György Károlyi (karolyi@reak.bme.hu)

**Abstract.** We develop a conceptual coupled atmosphere–phytoplankton model by combining the Lorenz'84 general circulation model and the logistic population growth model under the condition of a climate change due to a linear time dependence of the strength of anthropogenic atmospheric forcing. The following types of couplings are taken into account: a) the temperature modifies the total biomass of phytoplankton via the carrying capacity, b) the extraction of carbon dioxide by phytoplankton slows down the speed of climate change, c) the strength of mixing/turbulence in the oceanic mixing layer is in correlation with phytoplankton productivity. We carry out an ensemble approach (in the spirit of the theory of snapshot attractors) and concentrate on the trends of the average phytoplankton concentration and average temperature contrast between the pole and equator, forcing the atmospheric dynamics. The effect of turbulence is found to have the strongest influence on these trends. Our results show that when mixing has sufficiently strong coupling to production, mixing is able to force the typical phytoplankton concentration to always decay globally in time and the temperature contrast to decrease faster than what follows from direct anthropogenic influences. Simple relations found for the trends without this coupling do, however, remain valid, just the coefficients become dependent on the strength of coupling with oceanic mixing. In particular, the phytoplankton concentration and its coupling to climate is found to modify the trend of global warming, and is able to make it stronger than what it would be without biomass.

## 1 Introduction

Large scale general circulation models typically take into account the interaction of the atmosphere with land vegetation and marine biomass production in the form of a huge number of parametrized processes (see e.g. Marinov et al, 2010; Zhong et al, 2011; Mongwe et al, 2018; Wilson et al, 2018). A basic understanding of such coupling is, however, easier to obtain in low

order conceptual models, where even analytic results may be available. Probably the most important component that needs to be included in such conceptual models is the phytoplankton content of the ocean.

Oceans are the major sink for the atmospheric $CO_2$ (Hader et al, 2014; Li et al, 2012). $CO_2$ is either stored as dissolved inorganic carbon or transferred to the underlying sediment by biological carbon pump. The motor of the biological pump is phytoplankton which is one of the major components of the global carbon cycle hence influencing decisively atmohperic $CO_2$ (Hutchins and Fu, 2017; Sanders et al, 2014; Turner, 2015; Falkowski et al, 2000). Besides, phytoplankton is resonsible for nearly half of the total primary production on Earth (Basu and Mackey, 2018). Consequently, it is extremely important to understand the intercation of primary production in oceans with effects contributing to global climate change. However, the task is very challenging: change in atmospheric $CO_2$ level can have opposite impact on processes influencing the phytoplankton and the intensity of the biological carbon pump. Increased atmospheric $CO_2$ level increases ocean temperature, decreases pH, increases water stratification, influences general oceanic circulation. These can all modify the net productivity and the composition of phytoplankton, and can have either positive or negative net effect on the biological carbon pump (Basu and Mackey, 2018, and references therein).

In spite of the current trend to include biogeochemistry in climate models (see e.g. Schlunegger et al, 2019), a basic understanding of such processes is still limited. It is still under debate whether net primary production is increasing or decreasing in coupled carbon–climate models as a consequence of warming induces production increase and stronger nutrient limitations induced by increased stratification (Laufkötter et al, 2015). The situation appears to be similar to the understanding of thermal or fluid dynamical concepts decades ago. The study of e.g. the energy balance Ghil (1976) or of the thermohalin ciculation Stommel (1961) started with elementary conceptual models which later evolved into more complex ones, and are by now decisive components of cutting-edge climate models. We therefore propose here to study a conceptual atmosphere–phytoplankton model where emphasis is on a proper choice of couplings (feedbacks). Thus, in our model, an increase of the global temperature affects the global primary production of ocean. As we emphasize above, phytoplankton plays significant role in the global $CO_2$ balance (De La Rocha and Passow, 2014; Falkowski, 2014; Guidi et al, 2016), hence our aim is to take an elementary description of phytoplankton dynamics coupled to an elementary model of the atmosphere. The direct effect of increased $CO_2$ concentration on phytplankton dynamics can be stimulating or inhibiting, we study both scenarios. As atmospheric model, we use Lorenz' elementary global circulation model (Lorenz, 1984), which was extended to mimic climate change (Drótos et al, 2015). The global phytoplankton concentration is represented by a simple logistic model in which the carrying capacity is coupled with the $CO_2$ content (direct effect) depending also on the concentration itself and on the wind energy influencing the oceanic mixing layer (indirect effect of climate change).

An appropriate treatment of even elementary models describing climate change is not obvious since basic parameters change with time and, therefore, traditional long-time averages cannot be used to define (in the sense of any statistical quantifiers) a *state of the climate*. An emerging new view, already embraced by Drótos et al (2015), follows a different route to obtain information on instantaneous statistical quantifiers (e.g. expected, average properties) of the climate. Since our information on the actual state of the climate is incomplete, one imagines an ensemble of parallel Earth systems carrying parallel climate realizations subjected to the *same set of physical laws*, boundary conditions and external forcing, but with *different initial*

*conditions*. Then the chaotic or turbulence-like properties of the climate dynamics allows for distinct climate realizations (for a review see Tél et al (2019)). These realizations, however, cannot be arbitrary since only those are permitted that are compatible with physical laws and the given forcing. The ensemble of realizations defines a probability distribution of all the relevant variables at *any instant of time* from which one can obtain expected, ensemble average properties of the climate (for more details, and mathematical aspects, see Sec. 5).

It is therefore natural to use the ensemble view in our conceptual biogeochemistry model, too. The ensemble approach in it corresponds to generating parallel atmosphere–phytoplankton realizations from different initial conditions. In our model, the number of variables is 4, hence the snapshot attractor in the full state space is difficult to visualize. We therefore concentrate on ensemble averages, and the internal variability will be expressed in terms of variances. We include, in a simple, heuristic form, *important feedbacks* in the model: a) the change in the atmospheric temperature modifies phytoplankton concentration, b) the extraction of $CO_2$ by phytoplankton, and c) wind energy enhances the strength of turbulence in the oceanic mixing layer which increases the phytoplankton production (Estrada and Berdalet, 1997; Peters and Marrasé, 2000; Jäger et al, 2010).

The paper is organized as follows. In section 2 we describe the model and define the relevant coupling parameters. Without mixing, exact relations can be derived, the most important of these are summarized in Section 3, while details of the calculations are relegated to Supplementary Material I. In the presence of mixing, numerical simulations are carried out in the spirit of snapshot attractors. The results are summarized in Section 4 where one learns that the extraction effect of $CO_2$ has the least influence on the general behavior in the presence of mixing. The feedback of the temperature contrast on the phytoplankton concentration has important consequences, but these are suppressed by a sufficiently strong mixing, which converts the typical phytoplankton concentration to always decay in time, and surprisingly, the typical temperature contrast is found to decrease faster than that solely by direct anthropogenic effects. Planar sections of the 4-dimensional snapshot attractor underlying the dynamics are presented in Section 5, and our conclusions are drawn in Section 6. Additional figures are presented in Supplementary Material II. A list of variables and parameters is given in Supplementary Material III, while Supplementary Material IV contains a sample of the C code applied during numerical simulations.

## 2  The model

The physical content of Lorenz's atmospheric circulation model for the midlatitudes (Lorenz, 1984, 1990) on one hemisphere is the following. The main forcing is the temperature difference $T_e - T_p$ between the Equator and the Pole. This is proportional to model variable $F$ influencing most directly the wind speed of the Westerlies represented by $x$. As an effect of baroclinic instability, cyclonic activity facilitates poleward heat transport, two modes of which are represented by $y$ and $z$. The model reads as follows:

$$\dot{x} = -y^2 - z^2 - ax + aF(t), \tag{1a}$$

$$\dot{y} = xy - bxz - y + G, \tag{1b}$$

$$\dot{z} = xz + bxy - z. \tag{1c}$$

For the parameter setting we take the common choice: $a = 1/4$, $b = 4$, $G = 1$. The equations appear in a dimensionless form with the time unit corresponding to 5 days.

By using time-dependent forcing, $F(t)$, as Drótos et al (2015), we also model the contribution of the varying $CO_2$ content in association with the greenhouse effect. Besides the variation of $CO_2$ due to effects appearing in $F(t)$, the extraction of $CO_2$ by phytoplankton is also included into our model. The $CO_2$ content stored in marine ecosystems, or buried in the sea bed, is correlated with primary production (Falkowski et al, 1998, 2003). Thus, as discussed in the Introduction, modelling the interaction of phytoplankton and atmospheric dynamics is a good proxy for studying marine ecosystem interaction with atmospheric dynamics. Hence we couple the Lorenzian atmospheric dynamics to that of the photosynthetizing oceanic biomass, assumed to be dominated by phytoplankton of concentration $c(t)$. The temperature contrast parameter thus also depends on the global phytoplankton concentration $c$: $F(t) \rightarrow F(c(t),t)$, with a form to be given below.

Spatial inhomogeneties in nutrient and consequently phytoplankton content due to e.g. oceanic eddies and upwellings are known to play an important local role in Nature. However, a global atmospheric model like (1) can adequately be coupled only to a global phytoplankton dynamics model. Therefore, the concentration itself is assumed in this simple set-up to follow a logistic population growth

$$\dot{c} = rc \left( 1 - \frac{c}{K(t)} \right). \tag{2}$$

Carrying capacity $K$ is taken to depend on the average temperature of the hemisphere, or, equivalently, on the temperature contrast $F$. As a consequence, $K$ depends on time also via the concentration $c$: $K(t) = K(c(t),t)$. We shall see that an important oceanic effect, that of the turbulence in the mixing layer, can be incorporated into carrying capacity $K$, although only on a global scale. Parameter $r$ sets the growth rate of the phytoplankton. If, e.g., $r = 1$, the phytoplankton characteristic time is $5/r = 5$ days, as that of the atmosphere. This latter choice will be kept throughout the paper. The assumption of Eq. (2) for the global phytoplankton dynamics tacitly implies that phytoplanton biomass determines the total biomass of the oceans, and also that no catastrophic events (no mass extinction or invasion of species) can take place in this model.

A basic feature of the observed climate change on Earth is that the polar temperature $T_p(t)$ increases, while the equatorial one $T_e$ remains practically constant (Serezze and Francis, 2006; Blunden and Arndt, 2013). We can thus write in suitable units the temperature contrast parameter as $F(t) = T_e - T_p(t)$. The mean temperature in these units is then $T(t) = (T_e + T_p(t))/2 = T_e - F(t)/2$. We are interested in dominant, leading order effects and assume therefore the carrying capacity to be coupled linearly by a small coupling constant to the mean temperature relative to some reference state of mean temperature $T_r$ (in which the temperature contrast parameter is $F_r = 2(T_e - T_r)$). The temperature difference $T - T_r$ is then $-(F - F_r)/2$. We therefore write

$$K(t) = K_r - \alpha(F(c(t),t) - F_r), \tag{3}$$

where $K_r$ is the carrying capacity in the reference state characterized by $F_r$. Coefficient $\alpha$ represents coupling a) between the carrying capacity $K$ and climate change, represented by $F$, and shall be called the *enrichment parameter*, see Fig. 1 where the full set of feedbacks considered in the model is schematically presented. This coupling may be either positive or negative. For

example, increased $CO_2$ level enhances the efficiency of photosynthesis ($\alpha > 0$), however, acidification because of increased $CO_2$ levels depresses respiration ($\alpha < 0$) (Reid et al, 2009; Mackey et al, 2015). Similarly, increased water temperature can have both positive and negative effect on phytoplankton biomass in different regions of Earth (Chust et al, 2014; Roberts et al, 2017). We shall, therefore, allow for both positive and negative values of $\alpha$ with $|\alpha|$ small.

Plankton dynamics influences the temperature contrast. If concentration $c$ increases, the temperature contrast $F$ increases, 125 too, because the biomass extracts more $CO_2$. In leading order, we therefore express the concentration-dependent temperature contrast parameter as a linear function of the concentration:

$$F(t) = F(c(t), t) = \beta(c(t) - c_r) + F_0(t) \tag{4}$$

with a small $\beta > 0$, where $c_r$ is the phytoplankton concentration in the reference state. Coefficient $\beta$ represents coupling b) due to the extraction of $CO_2$ by phytoplankton, and we therefore call $\beta$ the *extraction parameter* (see Fig. 1). The second term, 130 $F_0(t)$, represents the primary external forcing due to the $CO_2$ content of anthropogenic origin. The increase of both $F_0$ and $(c(t) - c_r)$ leads to an increase in the temperature contrast $F(t)$. With this form of $F$ the carrying capacity (3) is

$$K(t) = K_r - \alpha[\beta(c(t) - c_r) + F_0(t) - F_r]. \tag{5}$$

Without a restriction of generality, we can choose the reference carrying capacity to be $K_r = 1$, implying a reference concentration $c_r = 1$. This choice only rescales parameters $\alpha$ and $\beta$ in (3) and (4), respectively.

Starting from negative times, we assume the Earth system to be in climatic and population dynamical equilibrium up to time $t = 0$. This state, chosen as the reference state, is characterized by a time independent mean temperature $T_r$, concentration $c_r = K_r$, and $F_0(t) = F_r$. At time zero, climate change sets in expressed by a linear decrease in the primary temperature

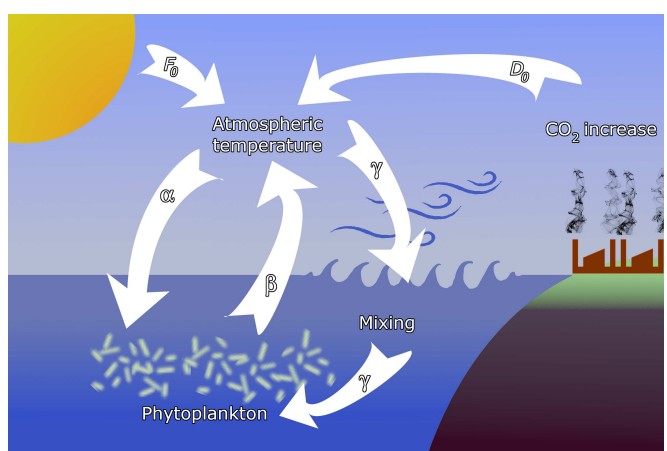

**Figure 1.** Sketch of the feedbacks considered in the model. Temperture contrast $F_0$ between the Pole and the Equator, containing also seasonal variabiliy, is augmented by anthropogenic effects $D_0$. The main interactions are $\alpha$) the effect of atmospheric temperature on biomass, $\beta$) the extraction of $CO_2$ by phytoplankton, and $\gamma$) oceanix mixing, driven by atmospheric dynamics, affecting phytoplankton productivity.

contrast:

$$F_0(t) = F_r - D_0 t, \tag{6}$$

expressing direct anthropogenic effects, with a decrease parameter $D_0 = 2/7300$ for $t > 0$ (Drótos et al, 2015). Since one year corresponds to 73 time units (365 days), $1y = 73$, this form expresses that the temperature contrast decreases by 2 units over 100 years. We shall take $F_r = 9.5$, with which the temperature contrast would go down, after a climate change period of 150 years, and without any change in the biomass concentration, to 6.5. We stop the climate change scenario in year 150 because model (1) loses its global chaotic property , which is a prerequisite even for a minimal climate model, for small $F$.

With this scenario (6) of the anthropogenic influence, the carrying capacity $K(t)$ is, in rescaled units,

$$K(t) = 1 - \alpha[\beta(c(t) - 1) - D_0 t], \tag{7}$$

where $D_0 = 0$ for $t \leq 0$ and $D_0 = 2/7300 = 2.7 \cdot 10^{-4}$ for $t > 0$.

We can model seasonality, too, as Lorenz also did (Lorenz, 1990), by augmenting (6) with a periodic term:

$$F_0(t) = F_r - D_0 t + A \sin(\omega t). \tag{8}$$

His choice was $A = 2$ with $\omega = 2\pi/73$, which we shall adopt. Our climate change starts with year 0, and this year begins at the time instant $t = 0$. Note that this time instant belongs to an autumnal equinox according to (8), and, furthermore, $F_r - D_0 t$ can be considered as the annual mean temperature contrast. Any time $t \bmod 73 = 0$ coincides with other autumnal equinoxes, and results will be presented on this day of the year throughout the paper.

Up to this point, the atmospheric variables have not entered the concentration dynamics. Without the linear and constant terms (representing dissipation and forcing, respectively), Eqs. (1) would conserve the total kinetic energy

$$E = \dot{x}^2 + \dot{y}^2 + \dot{z}^2$$

of the atmosphere. From the point of view of the biomass, it is natural to assume that the activity of the atmosphere influences the ocean dynamics within its uppermost mixing layer (Sverdrup, 1953; Whitt et al, 2017), in particular, the strength of turbulence, and hence the depth of mixing layer. Note that component $\dot{x}^2$ represents the contribution of zonal winds to the total atmospheric energy, while $\dot{y}^2 + \dot{z}^2$ represents wind energy staming from cyclonic activity. The depth of the mixing layer, and consequently the carrying capacity, are assumed to increase linearly with $E$ in our model, with a small coupling constant. The most general form of the carrying capacity $K$ is thus

$$K(t) = 1 - \alpha[\beta(c(t) - 1) - D_0 t + A \sin(\omega t)] + \gamma(\dot{x}^2 + \dot{y}^2 + \dot{z}^2). \tag{9}$$

Here $0 \leq \gamma \leq 0.2$ is the strength of a weak coupling c) due to oceanic mixing what we call the (oceanic) *mixing parameter*. This provides a feedback between the phytoplankton dynamics and the climatic variables (see Fig. 1).

## 3 Analytic results without mixing

Without mixing ($\gamma = 0$), Eq. (2) can be solved by a simple ansatz of $c(t)$, irrespective of the atmospheric dynamics. This leads to analytic results concerning some properties of the model, which are summarized in Supplementary Material I. As an example, we give here two simple relations which help to understand the general tendencies of the system. Eq. (2) with (9) is shown for $\gamma = 0$ to possess linear behavior for long times, inherited from the temperature contrast of anthropogenic origin:

$$c(t) \sim St, \quad F(t) \sim -Dt. \tag{10}$$

Naively, one expects that an increased $CO_2$ level (smaller $F$ in (1)) leads to a higher carrying capacity and concentration of the phytoplankton, and a slower decrease of the temperature contrast, i.e., $S$ ($D$) should increase (decrease) with the enrichment parameter. However, only by calculating the precise dependence can reveal whether these trends are important or hardly discernible. The linear coefficient, slope $S$ in the phytoplankton concentration's time dependence is found to be

$$S = \frac{D_0 \alpha}{1 + \beta \alpha} \approx D_0 \alpha. \tag{11}$$

The approximate equality reflects that the product $\alpha \cdot \beta$ is quadratically small since both the enrichment parameter $\alpha$ and the extraction parameter $\beta$ are small quantities. Hence the leading order behavior in $\alpha$ is linear. This relation shows that for a positive (negative) coupling $\alpha$ the phytoplankton concentration increases (decreases) proportionally with the enrichment parameter $\alpha$, and with the slope $D_0$ of the anthropogenic temperature contrast.

The linear coefficient in the temperature contrast is

$$D = \frac{D_0}{1 + \beta \alpha} \approx D_0(1 - \beta \alpha). \tag{12}$$

The approximate equality provides, again, the leading order behavior in $\alpha$. The relation indicates that in the case of a positive enrichment parameter $\alpha$ the phytoplankton dynamics *weakens* the climate change, weakens the trend from $D_0$ to $D$ in the temperature contrast, as expected. Quite surprisingly, however, the effect is rather weak since $\alpha \cdot \beta$ is quadratically small. Relations (11,12) also suggest that the role of (a weak) extraction coupling is not essential: the leading behavior in $S$ is independent of $\beta$. Its effect is weak also in $D$, this quantity coincides with the anthropogenic slope $D_0$ for $\beta = 0$ (as also follows from (4)), it deviates from $D_0$ very little otherwise.

It is worth noting that relations (11,12) remain valid for the time-averaged trends in the presence of a seasonal periodicity, as also shown in Supplementary Material I. Relations (11,12) are independent of initial conditions, they represent the snapshot attractor of the problem projected on variable $c$. This attractor is fixed point-like, but changes in time (moves uniformly, or with an oscillation superimposed when seasonality is taken into account). There is, however, no internal variability in the concentration variable $c$, although an extended, fractal snapshot attractor underlies the atmospheric variables exactly as in the model of Drótos et al (2015) where no phytoplankton dynamics was taken into account.

## 4 Numerical results with mixing: trends in the fully coupled model

In the interesting case of nonnegligible mixing, no analytic result can be obtained. This implies a nontrivial biomass dynamics for $\gamma > 0$, a dynamics exhibiting internal variability in variable $c$, too. To explore this regime, we carried out a sequence of numerical simulations of the full 4-variable dynamics. The following parameters are kept fixed (as indicated in the previous section): $r = 1$, $D_0 = 2/7300$, $F_r = 9.5$, $A = 2$, $\omega = 2\pi/73$, and we vary $\alpha$, $\beta$, and $\gamma$. Equations (1) and (2) with (4), (8) and (9) are solved with the classical 4th-order Runge-Kutta method with a fixed time-step $dt = 0.01 \approx 1.37 \times 10^{-4}$ y.

To start with, Fig. 2a shows a few individual concentration realizations (colored lines) $c(t)$ for a mixing parameter $\gamma = 0.1$ ($\alpha = 0.05$, $\beta = 0.1$), along with the ensemble average $<c>(t)$ of 50 000 realizations initiated at $t = -20$ y (purple line). Here and in what follows angled brackets $<>$ will always denote averages taken with respect to our ensemble at a given time instant, $t$. The individual cases are all rather different. For $t < 0$ there is no climate change, nevertheless, the individual time series $c(t)$ exhibits strong variance, very similar to those observed for $t > 0$, i.e., they are unable to properly reflect the ensemble, and, in particular, the lack of climate change for $t < 0$. The ensemble average, $<c>(t)$, however, provides a plateau here up to $t = 0$ indicating clearly the stationarity of the climate and, therefore, of the biomass dynamics in this range. In Fig. 2b we display the source of the time variability of the phytoplankton concentration, the total kinetic energy $\dot{x}^2 + \dot{y}^2 + \dot{z}^2$ of the atmosphere at each time instant. The deviation of the individual ensemble member time series from the average is represented here by means of

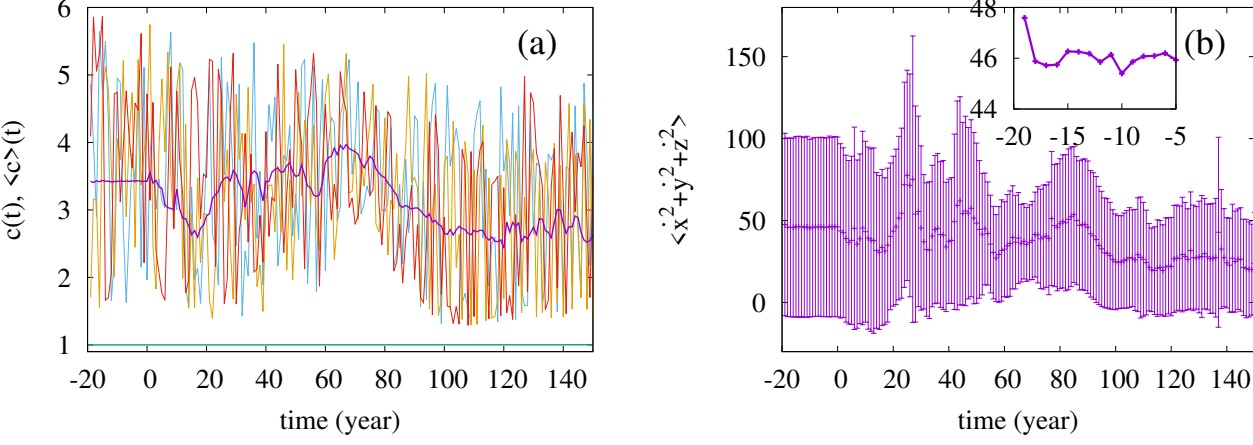

**Figure 2.** Ensemble properties before ($t < 0$) and after ($t > 0$) the onset of climate change. a) Phytoplankton concentration $c$ as a function of time for three random initial conditions in different colors for $\alpha = 0.05$, $\beta = 0.1$, $\gamma = 0.1$. The purple line is the ensemble average $<c>(t)$ for 50000 trajectories started with random initial positions in the range $x \in [-0.5, 3]$, $y, z \in [-2.5, 2.5]$, $c \in [0.9, 1.1]$ at year -20. The green line (close to $c = 1$) shows the expected phytoplankton concentration without any mixing ($\gamma = 0$) as predicted by Eq. (10). (The increase for $t > 0$ is so weak that one hardly recognizes it on this graph.) b) Time-dependence of the ensemble average (dark violet "+" marks) of the total atmospheric kinetic energy $<\dot{x}^2 + \dot{y}^2 + \dot{z}^2>$ for the same ensemble as the one used for $c$ in (a). Violet bars indicate the standard deviation. The inset shows the blow up of the initial part of the average in b).

the standard deviation evaluated over the ensemble (violet bars). The average kinetic energy, along with its ensemble variance, is also constant before the climate change and starts an irregular time dependence right after $t = 0$. One can observe that the kinetic energy strongly influences the phytoplankton concentration (via the carrying capacity $K$ in (9)), but the concentration itself contributes to the $CO_2$ content, and to the temperature contrast $F$, see (4), forcing the atmosphere (as will be demonstrated in Fig. 3). The feedback of the atmosphere on phytoplankton is rather strong in this set-up with $\gamma = 0.1$, also expressed by the strong difference between the green line (obtained for $\gamma = 0$) and the purple line in Fig. 2a illustrating that this coupling leads to an enormously enhanced biomass concentration.

It is visible in the inset to Fig. 2b that the ensemble average curve shows some change during the first 5 years (between $t = -20$ and -15 y). This indicates (along with several other simulations, not shown) that the convergence to the snapshot attractor takes about $t_c = 5$ y. The numerical data after $t = -15$ y thus represent parallel atmosphere-phytoplankton realizations on the snapshot attractor of the system.

The considerable deviation of the individual time series from the ensemble average indicates that the formers are not representing properly the mean climate state, as also pointed out by Drótos et al (2015). Therefore, from here on, we shall concentrate on ensemble averages, and consider the variance about these as a measure of the internal variability (the size of the snapshot attractor in the chosen variable).

We carried out similar simulations with other extraction parameter values from the range $\beta \in [0.0, 0.5]$ and found that $\beta$ does not have much effect on the average phytoplankton concentration, the curves for various values of $\beta$ are close to each other (see Fig. S1 in Supplementary Material II). In what follows, therefore, we stick to a single value, $\beta = 0.1$.

The time-dependence of the typical (ensemble averaged) temperature contrast $<F>(t)$ forcing the atmospheric variables in (1) is shown in Fig. 3. The value of $<F>(t)$ at each time instant is computed from Eqs. (4), (8), with the average values $\langle c \rangle$ (ensemble average over 50000 trajectories at that time instant) in place of $c$. The fluctuations in the $<F>(t) = F(<c>(t), t)$ curve of Fig. 3 follow the fluctuations in the average phytoplankton concentration, but, for small values of $\gamma$, the linear decrease of $F(t)$ is recovered. In other words, for weak mixing (small values of $\gamma$) the trend in the forcing $F(t)$ follows quite closely the direct anthropogenic trends. For strong mixing ($\gamma \leq 0.1$), however, the fluctuations have longer time-scale, hence the trends imposed by anthropogenic effects are less obvious, in particular, on shorter time-scales. A comparison of Figs. 3 a and b belonging to $\alpha = 0.05$ and $\alpha = -0.05$, respectively indicates that a change in the sign of the enrichment parameter leads to only minor differences in the general trends.

Next, we study the dependence of the ensemble average of the phytoplankton concentration on the strength of mixing. We have seen in Fig. 2 that for $\gamma = 0.1$ strong deviations appear from the trend, $\alpha D_0$, occurring without mixing. The time dependence of $<c>$ for mixing parameters on this order of magnitude, shown in Fig. S2 of Supplementary Material II, confirms the existence of large fluctuations. The time dependence of $<c>$ for much smaller values of $\gamma$ are shown in Fig. 4. The linearly increasing trend in harmony with (10) and (11) gradually disappears, and large scale fluctuations are visible even for $\gamma = 0.005$.

It seems that even a small coupling of the atmospheric variables $x$, $y$ and $z$ to the phytoplankton dynamics will result in large variations of $<c>$ and in the suppression of the anthropogenic trends on short terms. One can also conclude from these

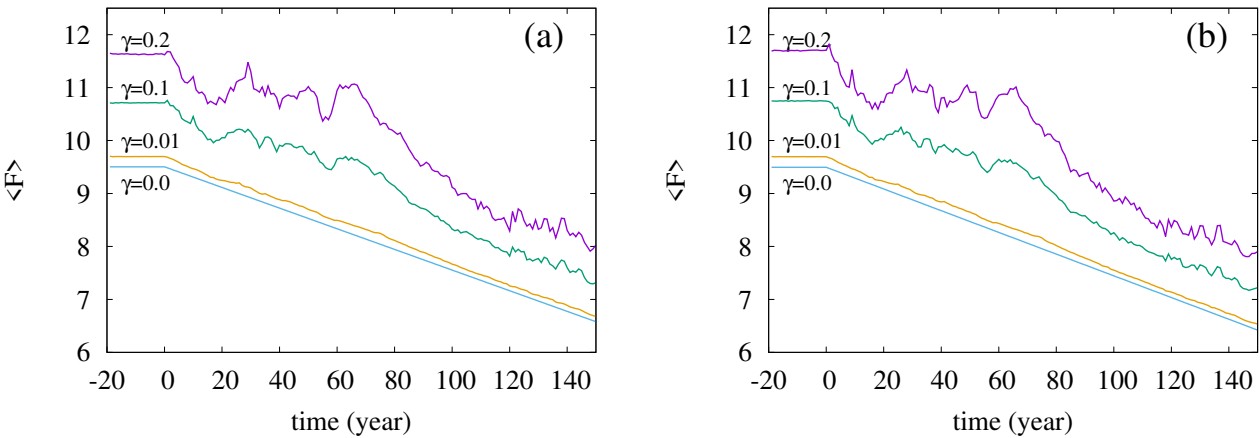

**Figure 3.** Time-dependence of the ensemble averaged atmospheric forcing $<F>(t) = F(<c>(t),t)$ in case of $\gamma = 0, 0.01, 0.1$ and $0.2$ for a) $\alpha = 0.05$, b) $\alpha = -0.05$. The slope of the blue line for $\gamma = 0$ corresponds to $D$ in (12).

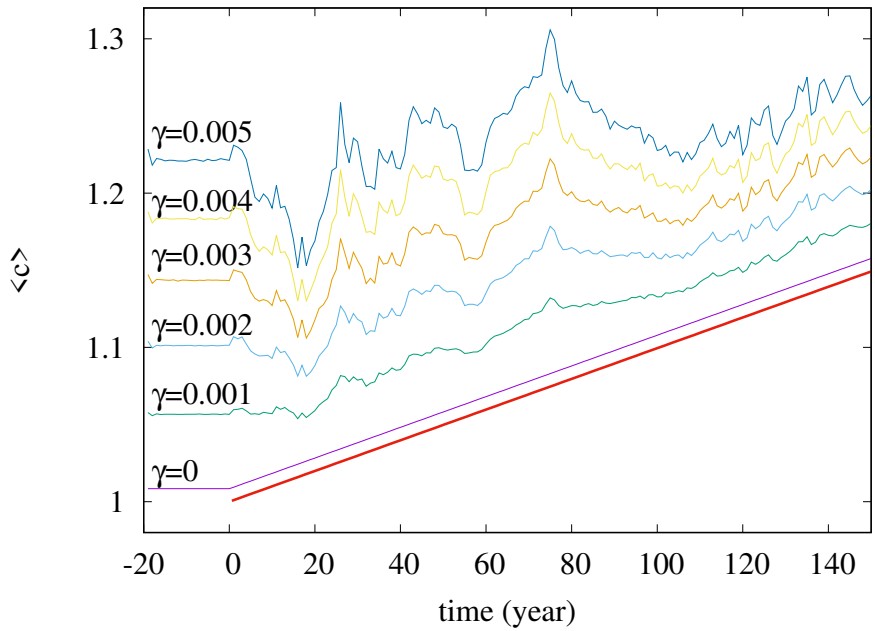

**Figure 4.** Ensemble averaged phytoplankton concentration $\langle c \rangle$ as a function of time for $\alpha = 0.05$ for various values of $\gamma$ ($\gamma = 0, 0.001, 0.002, 0.003, 0.004$ and $\gamma = 0.005$). The thick red line shows the expected phytoplankton concentration in lack of mixing ($\gamma = 0$) as predicted by Eq. (11).

figures that a coupling with $\gamma \geq 0.002$ should already be considered strong in the atmosphere-ocean interaction, at least from the point of view of the phytoplankton dynamics.

Now we investigate the effect of the enrichment parameter $\alpha$ on the phytoplankton concentration. We have seen in Fig. 4 that for small $\alpha = 0.05$, the short term trends are destroyed for $\gamma > 0.002$. We see in Fig. 5 that with an increase of $|\alpha|$, a trend might reappear at even higher values of the mixing parameter $\gamma = 0.01$. Indeed, for $\alpha$ between roughly -0.05 and 0.05, no trend is visible, large scale fluctuations stamming from the internal variablility of the dynamics rule the behavior of the average phytoplankton population. For $|\alpha| \geq 0.1$, however, we see that trends emerge, there is an increasing trend for positive, and a decreasing trend for negative $\alpha$ with a slope similar to the one given by the analytic calculation valid for $\gamma = 0$.

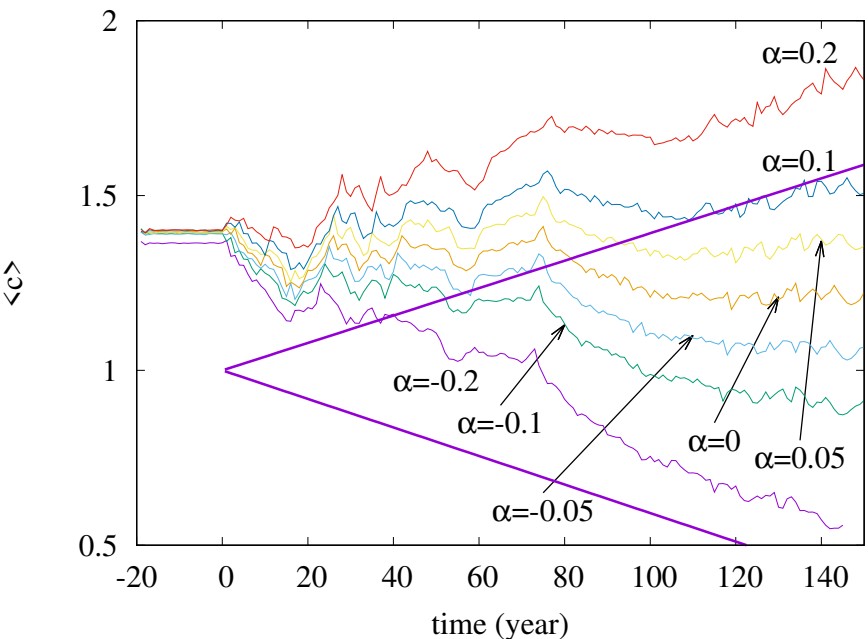

**Figure 5.** Average phytoplankton concentration $\langle c \rangle$ as a function of time for $\gamma = 0.01$ for various values of $\alpha$ (-0.2, -0.1, -0.05, 0.0, 0.05, 0.1 and 0.2). The thick purple lines show the expected phytoplankton concentration without mixing ($\gamma = 0$) as predicted by Eq. (11) for $\alpha = -0.2$ (lower line) and $\alpha = 0.2$ (upper line).

From the same set of $\alpha$ values used to construct Fig. 5, we show the time-dependence of the average forcing $< F > (t)$ for an intermediate ($\gamma = 0.01$) and a large ($\gamma = 0.1$) mixing parameter in Fig. 6a and b, respectively. Interestingly, for each value of $\alpha$ and $\gamma$, the $< F > (t)$ graphs show a nearly linear decay, the slope depending somewhat on $\alpha$. It seems that the direct anthropogenic component is dominant in the average forcing term, in particular for $\gamma = 0.01$, but this also holds qualitatively for $\gamma = 0.1$ (see Fig. 6b). We thus conclude that a mixing parameter on the order of $0.1$ is not yet strong from the point of view of the forcing. This is in harmony with the observation that the atmospheric kinetic energy hardly depends on the mixing strength (see Fig. S3 of Supplementary Material II): the atmosphere is rather resistant against the feedback from the biomass. Although an increased (decreased) amount of phytoplankton present in the system results in an increased (decreased) temperature contrast and hence in a decreased (enhanced) climate change, this effect is quite small. The order of magnitude of

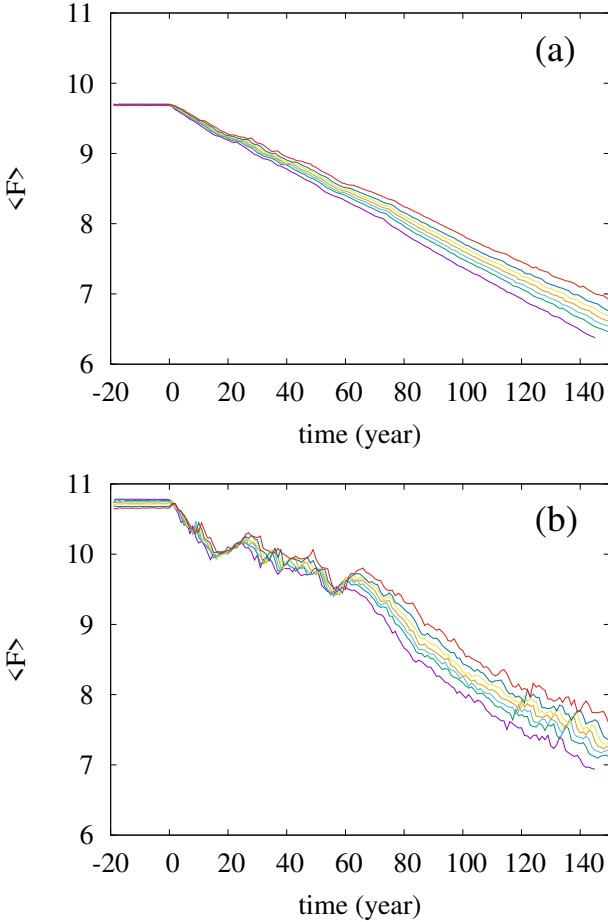

**Figure 6.** Time dependence of the average forcing $< F > (t)$ in case of (a) $\gamma = 0.01$ and (b) $\gamma = 0.1$. The values of the enrichment parameter from top to bottom are $\alpha = 0.2, 0.1, 0.05, 0, -0.05, -0.1$ and -0.2.

the effect of the phytoplankton concentration on $< F > (t)$ can be assessed by observing in Fig. 5 that $(\langle c \rangle - 1)$ falls between -0.5 and 1 at $t = 150$y. Multiplied by our fixed $\beta = 0.1$, as (4) requires, one finds a range of 0.15, which is much smaller than the final value of $F$, about 7, at 150 y. This is comparable with the spread of the temperature contrast at the end of year 150 in Fig. 6a and b. Note that these conclusions are drawn from the average temperature contrast. No trend can be extracted if instantaneous values of a single simulation are used instead of the ensemble average, in the same spirit as in Fig. 2a.

Next, we study quantitavely how the trend observed in the ensemble average of the phytoplankton concentration changes with the parameters. To this end, we fit a straight line to the time dependence of the ensemble average $\langle c \rangle (t)$ of the phytoplankton concentration for $t > 0$ for various values of parameters $\alpha$ and $\gamma$. The slope $S(\alpha, \gamma)$ of the best fit line in the presence of mixing gives information on the trend of the phytoplankton concentration, that is, on how quickly the concentration changes with time on (ensemble) average. We have also computed the standard deviations of this fit from the measured values to gain information

on the fluctuations appearing in individual members of the ensemble. We found (not shown) that in case of a strong trend (slope
of time-dependence far from zero) we find small fluctuations and vice versa.

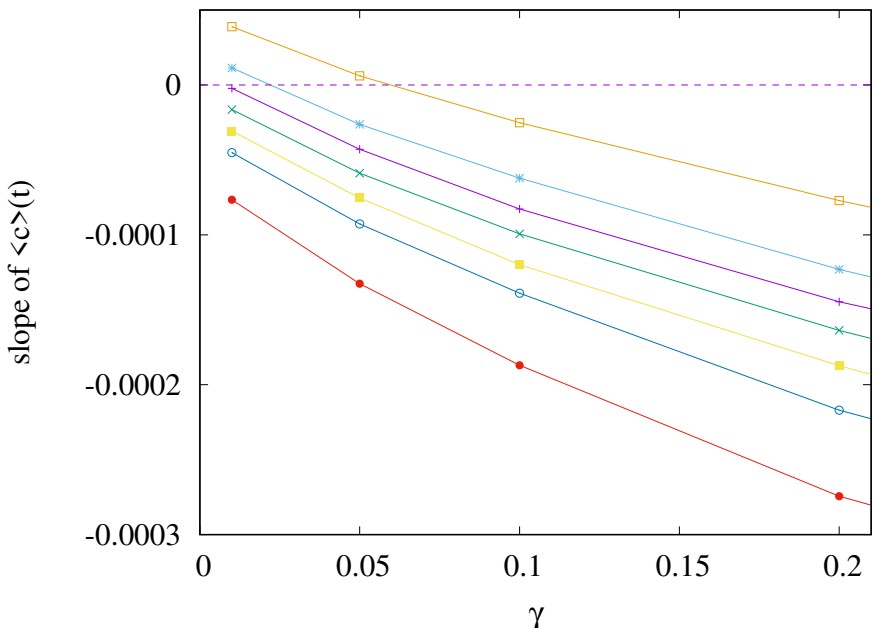

**Figure 7.** Slope $S$ of the ensemble average of the phytoplankton concentration, $<c>$, for $t>0$ as $\gamma$ is varied, data shown for various values of $\alpha$, from top to bottom, $\alpha = 0.2, 0.1, 0.05, 0, -0.05, -0.1$ and -0.2.

In Fig. 7 we show the approximate slope $S(\alpha, \gamma)$ of the $\langle c \rangle(t)$ curves as a function of the mixing parameter $\gamma$. We see that the measured slopes, that is, the trends in the time-dependence, decrease with increasing values of $\gamma$. We also found that the fluctuations (not shown) are enhanced when $\gamma$ increases. This implies that when mixing becomes stronger, not only the phytoplankton concentration is decreasing for any $\alpha$ (the slope is negative), but even drops faster (the slope is decreasing).
Note that the initial concentration from which the decrease starts at $t=0$ is higher for larger $\gamma$ (stronger mixing), see Fig. 4 and Fig. S2 of Supplementary Material II. Concerning the fluctuations, we call the attention to the fact that in nearly all figures exhibiting time dependence one can observe a decrease in the amplitude of variations for longer times, for $t > 100\text{y}$ approximately. This appears to be a consequence of the decrease of the total atmospheric kinetic energy with time, due to the overall decrease of the temperature contrast in time, as Fig. S3 of Supplementary Material II. also illustrates. At a fixed mixing
parameter $\gamma$, the strength of mixing is proportional to the kinetic energy, which is thus decreasing in time. Since the carrying capacity is assumed to linearly depend on the kinetic energy (see (9)), $K$ also decreases in time. Thus, the phytoplankton concentration and its fluctuations are also decreasing with time.

It is worth also noting that even if for $\gamma = 0$ the trend in $<c>$ would be increasing for positive enrichment parameters, see (11), it is the increase of $\gamma$ that converts all trends to be negative. It remains true, however, that the trend for a positive $\alpha$ is less

negative than for a negative $\alpha$. In other words, for sufficiently strong mixing, the phytoplankton concentration always decreases with time due to climate change, the sign of the enrichment parameter only influences the strength of decrease.

If we plot the same data shown in Fig. 7 as a function of $\alpha$ instead of $\gamma$, see Fig. 8a, we see that the increase in the enrichment parameter increases the trend in the phytoplankton concentration. It is a surprising observation that even if the change in the mixing parameter changes the slopes essentially, their $\alpha$-dependence remains similarly linear as for $\gamma = 0$ given
in (11). Plotting the slope $-D(\alpha, \gamma)$ of the time-dependent ensemble averaged forcing $< F >$ as a function of $\alpha$, see Fig. 8b, a very weak dependence is found (note the vertical scale). On a closer look, the $\alpha$-dependence is linear, and is increasing. This is in harmony with the expectation that the $CO_2$ extraction is weaker when the phytoplankton concentration is lower. With the exception of small $\gamma$ values, the slopes are more negative than the direct anthropogenic one, $-D_0$. It is a remarkable finding suported by our results that a large mixing parameter enhances the speed of the climate change, irrespective of the sign of the
enrichment parameter.

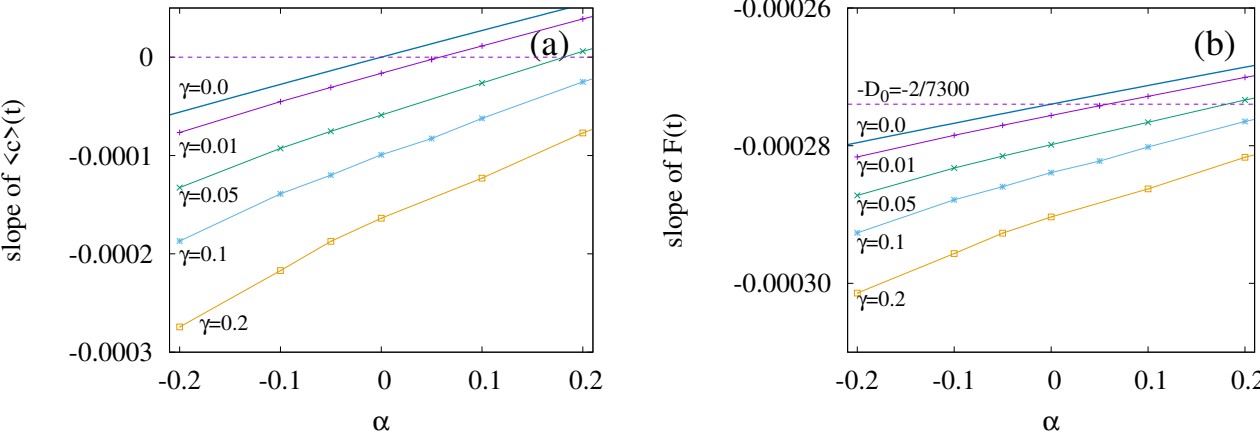

**Figure 8.** Slope (a) $S$ of the ensemble averaged phytoplankton concentration $< c > (t)$ and (b) $-D$ of the average forcing $< F >$ for $t > 0$ as $\alpha$ is varied, data shown for various values of $\gamma$. The $\gamma = 0$ curve shows the $\alpha$-dependence of (a) $S$ and (b) $-D$ from Eqs. (11,12). Dashed lines mark the slopes for $\alpha = 0, \gamma = 0$.

We see that the trends predicted by Eqs. (11,12) are approached when $\gamma$ is decreased. What is even more interesting, the dependence of the trends on $\alpha$ remains the same for any $\gamma$. In particular, we find a numerical fit of the slope $S$ of $< c > (t)$ for $\beta = 0.1$ as

$$S(\alpha, \gamma) = \alpha D_0 (1 + 3.8\gamma) - 2D_0 \gamma^{0.75}. \tag{13}$$

A similar expression is obtained from the slopes of the averaged forcing $< F > (t)$ that replaces $D_0$ found in (12) for $\gamma = 0$ by

$$D(\alpha, \gamma) = D_0 [1 - \alpha\beta(1 + 3.8\gamma)] + 2\beta D_0 \gamma^{0.75}. \tag{14}$$

It is surprising that the leading order linear behavior in the enrichment parameter $\alpha$ found for $S$ and $D$ without any mixing remains valid for practically the entire $\gamma$ range investigated, just the coefficients become $\gamma$-dependent.

## 5 Snapshot attractors

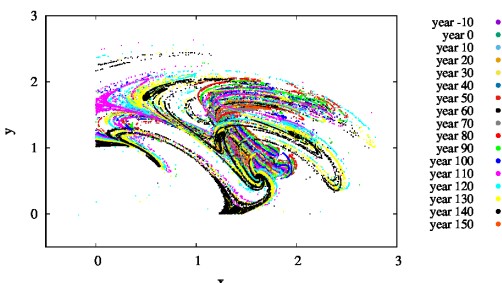

**Figure 9.** The projection of the $z = 0$, $\dot{z} > 0$ section of the snapshot attractors on the $x, y$ plane for $\beta = 0.1$, $\alpha = 0.05$, $\gamma = 0.1$. The snapshot attractors at intervals of 10 years are shown with purple ($t = -10$ y), green ($t = 0$ y), cyan ($t = 10$ y), light orange (($t = 20$ y), yellow ($t = 30$ y), dark cyan ($t = 40$ y), dark red ($t = 50$ y), dark grey ($t = 60$ y), grey ($t = 70$ y), red ($t = 80$ y), light green ($t = 90$ y), blue ($t = 100$ y), pink ($t = 110$ y), light blue ($t = 120$ y), bright yellow ($t = 130$ y), black ($t = 140$ y), dark orange ($t = 150$ y). They are generated by initiating $7 \times 10^7$ random initial conditions at year -20.

The mathematical concepts underlying the ensemble view are snapshot (Romeiras et al, 1990) or pullback (Ghil et al, 2008) attractors. One might consider the ensemble of all permitted climate realizations over all times as the pullback attractor of the problem, and the set of the permitted states of the climate at a given time instant as the snapshot attractor belonging to that time instant (their union over all time instants is the pullback attractor). Both views express that the climate system possesses a plethora of possibilities. In the terminology of climate science, climate has a strong internal variability (e.g.Stocker et al (2013)). The concept of snapshot or pullback attractors is nothing but a reformulation of this fact in dynamical terms.

In numerical simulations, we consider the members of an ensemble simulation to describe parallel climate realizations only after the initial conditions are "forgotten", transient dynamics disappears. Due to dissipation, this time is typically short compared to the time span of interest. Such an ensemble approach was shown to be the only method providing reliable statistical predictions in systems with underlying nonpredictable dynamics (since in this class the traditional approach based on single time series is known to provide seriously biased results). A number of papers illustrate these statements within the physics literature (see. e.g. (Romeiras et al, 1990; Lai , 1999; Serquina et al, 2008)), as well as in low order climate models (Chekroun et al, 2011; Bódai et al, 2011; Bódai and Tél, 2012; Bódai et al, 2013; Drótos et al, 2015), in general circulation models (Haszpra and Herein, 2019; Kaszás et al, 2019; Pierini et al, 2018, 2016; Drótos et al, 2017; Herein et al, 2017; Bódai et al, 2020; Haszpra et al, 2020; Haszpra and Herein and Bódai, 2020) and also in experimental situations (Vincze, 2016; Vincze et al, 2017).

For several parameter values, we also determined the snapshot attractors of the coupled model. An example is given in Fig. 9 where we see the attractor on the $z = 0$ slice of the atmospheric dynamics with the corresponding $c$-values not shown directly. Different colors indicate different time instances separated by 10 y, clearly indicating that the attractor is changing in time. As the colors indicate, the projection to the $(x, y)$ plane of the $z = 0$ cross-section of the snapshot attractor has a minimum

size in years 60–80, after which it increases again, and the maximum extension is reached by about year 150. Note that one
cannot decide how much of the time dependence is a consequence of $F_0(t)$ or of the phytoplankton concentration. Due to the
couplings between the biomass and the atmosphere, the direct anthropogenic effect cannot be separated from the effect of the
biomass.

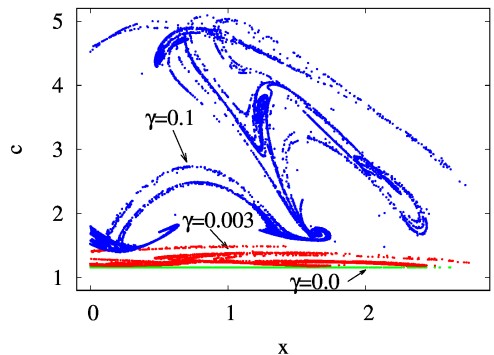

**Figure 10.** The projection of the $z = 0$, $\dot{z} > 0$ section of the snapshot attractors at year 150 on the $x, c$ plane for $\beta = 0.1$, $\alpha = 0.05$, and for
$\gamma = 0.0$ (green), 0.003 (red) and 0.1 (blue).

By investigating a projection of the snapshot attractor on a plane containing concentration $c$ as one of the axes, the influence
of mixing on the internal variability within $c$ can be visualized. In Fig. 10, the $z = 0$ slice of the snapshot attractor of a given
time instant is shown for three values of $\gamma$, projected to the $x, c$ plane, that is, the $y$ values are not shown. We see that the
extension of the snapshot attractor in the $c$ direction is greatly affected by the strength of mixing: the $c$ extension is zero
for $\gamma = 0$, but increases rapidly for increasing $\gamma$. Parallel to this, the pattern becomes interwoven in the space of variables,
suggesting that the $c$-dynamics becomes more and more complex in time, too. It is the increasing size and complexity of the
snapshot attractor in the $c$ direction which is reflected in the increase of the strength of fluctuations in Fig. 4 and Fig. S2 of
Supplementary Material II.
We also investigated the extremes of the snapshot attractors. That is, at a fixed time instant, we looked for those values of
e.g. $x$, for which only 10 % of values can be found on the snapshot attractor below (lower extreme) or above (higher extreme)
$x$. These values of $x$ are shown in Fig. 11a as a function of time. The interval between these thresholds is a measure of the size
of the extension of the snapshot attractor at a given time instant. Clearly, this size undergoes strong variations as a function of
time. The same is shown in Fig. 11b for the time dependence of the $c$ extension of the snapshot attractor: the upper (lower)
curve shows the value of $c$ above (below) which only 10 % of the values appear on the snapshot attractor. Again, we see
considerable variations in time. It is interesting to note that, as these figures indicate, there is no unique trend in the size of the
snapshot attractors, although trends can be seen in averages taken with respect to the ensemble designating the attractor itself,
like e.g. in $< c >$ or $< F >$.

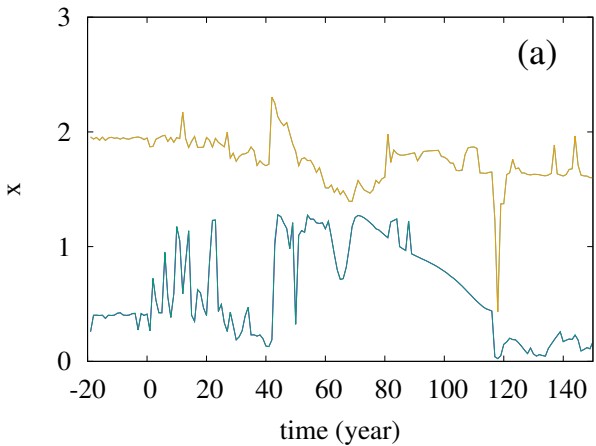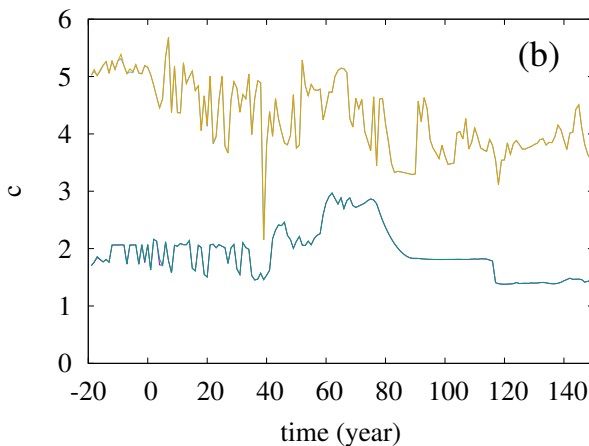

**Figure 11.** The extremes of the snapshot attractor for $\alpha = 0.05$, $\beta = 0.1$, $\gamma = 0.1$, Only 10 % of the points are found above (below) the higher (lower) values for each time instant for (a) $x$ and (b) $c$.

## 6  Conclusions

We have set up a conceptual coupled atmosphere–phytoplankton model by combining the Lorenz'84 general circulation model and the logistic equation under the condition of a climate change due to a linear decrease in the strength of direct anthropogenic forcing. The novel features of the model are in the choice of the possible forms of couplings. We allow for an influence of the biomass on the atmospheric forcing, modeling this way the extractions of $CO_2$ by phytoplankton, but the same forcing is able to modify the carrying capacity via its coupling to the temperature contrast characterized by the enrichment parameter. An additional atmosphere-ocean coupling is also taken into account mimicking the enhancement of phytoplankton primary production via increased atmospheric activity, i.e., via turbulent mixing. Our intention has been to include leading order effects, and hence the couplig constants are chosen intentionally to be small. Nevertheless, interesting consequences are found.

By investigating the parameter dependence of the ensemble average of the atmospheric forcing and the phytoplankton content, we have shown that

- even without mixing, the phytoplankton biomass interacts with the atmospheric forcing, and the coupling between the phytoplankton concentration and the temperature might weaken or strengthen the anthropogenic warming trend, the increase or decrease of the phytoplankton biomass depends on the sign of the enrichment parameter. In this regime, analytic results are available, see Eqs. (10, 11 and 12).

- increased mixing parameter enhances the total phytoplankton population biomass. Stronger coupling may enhance fluctuation to a degree that the anthropogenic component practically disappears (Fig. 4 and Fig. S2 of Supplementary Material II.).

- in contrast, mixing appears to depress the trend of the extraction of $CO_2$ by phytoplankton, and may force the phytoplankton population to globally decrease in time (see Fig. 7), although starting from a higher initial level.

- the coupling of mixing with phytoplankton biomass has a much weaker effect on the atmospheric forcing (see Fig. 6), as it is minimally expected from a coupled atmosphere-phytoplankton model.

- despite the strong modifications due to mixing, the dependence of trends on the strength of the coupling between the phytoplankton concentration and the temperature (the enrichment parameter) remains practically the same as without mixing (see Fig. 8).

We have obtained these results in a conceptual coupled atmosphere–phytoplankton model which contains a tractable number of variables and parameters. To our knowledge, this is the first attempt to understand the general and robust features of the interplay between the atmosphere and the biosphere in a climate change framework. One of our main results is that an increase in the global temperature reduces mixing intensity, which is the leading factor in decreasing the total biomass of primary producers. Interestingly, this result is in concordance with numerous studies applying Earth System Models with vastly more detailed plankton models (Bopp et al, 2013; Fu et al, 2016; Kwiatkowski et al, 2019), although other works report different observations (Laufkötter et al, 2015; Flombaum et al, 2020).

As far as we know, our work is the first step in the direction of studying the feedbacks between the atmosphere and the biosphere by a simple conceptual model. Our conclusions are robust in a mathematical sense, meaning that small changes in our model (inclusion of noise, for example) will not alter our main findings since snapshot attractors are robust. As long as the addition of other interactions only provide a small perturbation, our conclusions remain valid. In general, it is an open question in complex nonlinear systems whether neglected couplings to other subsystems and other simplifications could cause qualitative change in the dynamical behavior of a model. However, we see two important reasons why we believe our model goes in the right direction. First, the trends we find in our model are in accordance with the trends observed in the majority of complex models as mentioned above. Second, we believe that in our model the origin of trends are more transparent than in more complex models where this origin can be hidden among the multitude of variables, feedbacks and interations. Our model is a conceptual model, and as such, both the biological and climate models are highly simplified. However, one can consider it as a starting modul of an extendable model system. On the one hand, more trophical levels and inorganic resources can be easily added to the biological side of our model, on the other hand, simple ocean circulation models can extend the climate side of our model in order to make a first step to build more complex coupled models (Daron and Stainforth, 2013). We think that mutual interactions and iterations between conceptual models and detailed Earth System Models (ESM) help to reveal the disctinction between relevant and less relevant mechanisms and feedbacks behind climate change. We expect deeper insight into these feedbacks by studying conceptual and ESMs parallelly in the future.

*Code availability.* The C language code applied during the simulations is included in the Supplementary Material.

*Author contributions.* Gy. Károlyi, I. Scheuring and T. Tél worked out the outline of the applied model, with a special contribution of I. Scheuring to the biological background, Gy. Károlyi and T. Tél carried out the analytical calculations, Gy. Károlyi developed the simulation code, Gy. Károlyi and R.D. Prokaj carried out the simulations, all authors contributed to the preparation of the manuscript.

*Competing interests.* The authors declare that they have no conflict of interest.

*Acknowledgements.* We are thankful for the useful comments to T. Bódai, G. Drótos and T. Haszpra. This work was supported by the National
Research, Development and Innovation Office of Hungary under grant K-125171. IS is supported by Hungary's Economic Development and Innovation Operative Program (GINOP 2.3.2-15-2016-00057). GyK is supported by grant BME FIKP-VÍZ of EMMI.

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
