# Peer review of "Climate change in a conceptual atmosphere–phytoplankton model"

_Earth System Dynamics, 2019_

## Referee Comment (RC1) · Anonymous Referee #1 · 17 Jan 2020

This work presents an analysis of the feedbacks between atmosphere and ocean life with simplified mathematical models and detailed mathematical analysis. The questions and science considered in the paper are of broad relevance to researchers across many slices of the life sciences.

Overall, the study is very good and offers broadly applicable insights relevant to the Earth Sciences. However, the paper does not sufficiently put the Earth Science relevant findings and broader implications front and center for ESD and its audience. The findings are there, but the paper (and especially the introduction and conclusion) would benefit from expansion in this direction. Overall, I suggest re-arranging the material to highlight broader relevance.

Major Comments:

[Figure]

1. As mentioned, a greater focus on the broader Earth science issues and relevance is needed for an ESD paper. This can likely be accomplished through changes to the introduction and discussion/conclusions. Questions I wish the paper had addressed are along the lines of: what do these findings mean for more complex, process-based Earth System Models? I wanted more than Lines 355-356, and I think more could be said.

1a. Section 3 is a good example of how the paper is heavily focused on the details of the math. There's good scientific insight there: Lines 179-181 "The relation indicates that in the case of a positive enrichment parameter $\alpha$ the phytoplankton dynamics weakens the climate change, weakens the trend from D0 to D in the temperature contrast, as expected. Quite surprisingly, however, the effect is rather weak since $\alpha$ Âů $\beta$ is quadratically small." Is there a way to make that point up front in this section, with fewer references to equations, and to move even more of the equations to the SI? Adjustments along these lines throughout would be beneficial to appeal to a broader audience of researchers.

2. One easy change would be to include a table of variable and parameter notations, the quantities each notation represents, and any assumed values or boundaries imposed on the variables/parameters (such as alpha). This could be included in the SI, but is important to include, given the number of variables, parameters, and values being considered.

3. Similarly, any kind of figure/model schematic illustrating the setup and feedbacks (and their notations where possible) would be beneficial in the main text Section 2.

Specific comments:

1. Lines 42-57: some of this text would be better suited in a methods section than the introduction.

2. Line 60: What is the relevance to this work that the ensemble approach has been

used in other adjacent but distinct studies that presumably consider different models of different variables?

3. Figure 8 : Could you add a colorbar rather than (or in addition to) writing it out in the caption?

4. Section 4: overall I like this section very much but please make explicit mention that angled brackets always correspond to ensemble average, for every variable, early in the section.

5. Please consider making code and possibly some archive of the ensembles you run available to support open-access, reproducible science.

---

## Referee Comment (RC2) · Anonymous Referee #2 · 26 Apr 2020

Review of "Climate change in a conceptual atmosphere–plankton model" by Kárloy et al. for ESDD

This manuscript describes a conceptual modelling study of how phytoplankton respond to changes in temperature, atmospheric CO2, and ocean mixing. In this study the authors attempt to use a relatively simple model that contains many assumptions to simulate phytoplankton growth as a function of changes in anthropogenic forcing. While I am not opposed to simple conceptual models that focus on key processes or dynamics, I feel that the conceptual model presented here makes many assumptions that are not well justified. In addition, many relevant dynamics seem to be left out, e.g., other bottom-up controls on phytoplankton growth such as light and nutrients like N, P, and Fe, as well as top-down controls like grazing and mortality. While, it may be possible

that global phytoplankton-climate dynamics and responses to forcing can be explained without such factors (although I doubt it), in order for me to have confidence in their model I would need to see better evidence that justifies the model simplifications and model parameterizations. Many other conceptual studies of phytoplankton use laboratory studies or observations to justify their model structure and parameterizations, surely this can be done here as well. In addition, and perhaps more importantly, for me to have confidence in the model there needs to be some validation, i.e., comparisons to actual data. I know that the goal is to simulate climate change and obviously one cannot validate future projections. However, it should still be possible to come up with a clever way to validate the model (or key underlying equations) using observations. There is also no real attempt to contrast the results of this study with other phytoplankton focused climate studies that have used ocean-only or Earth system models (e.g., see citations listed below). Without satisfactory justification and validation of the model and the results this is just a mathematical exercise that while interesting, leaves the reader wondering if it has meaning. Therefore, I must recommend that the manuscript be rejected.

Bopp, L. et al. Multiple stressors of ocean ecosystems in the 21st century: projections with CMIP5 models. Biogeosciences 10, 6225–6245 (2013).

Flombaum, P., Wang, W.-L., Primeau, F. W., & Martiny, A. C. (2020). Global picophytoplankton niche partitioning predicts overall positive response to ocean warming. Nature Geoscience, 13(2), 116–120. https://doi.org/10.1038/s41561-019-0524-2

Laufkötter, C., Vogt, M., Gruber, N., Aita-Noguchi, M., Aumont, O., Bopp, L., et al. (2015). Drivers and uncertainties of future global marine primary production in marine ecosystem models. Biogeosciences Discussions, 12(4), 3731–3824. https://doi.org/10.5194/bgd-12-3731-2015

Kwiatkowski, L., Aumont, O., & Bopp, L. (2018). Consistent trophic amplification of marine biomass declines under climate change. Global Change Biology, (July), 0–2.

https://doi.org/10.1111/gcb.14468

Kwiatkowski, L., Bopp, L., Aumont, O., Ciais, P., Cox, P. M., Laufkötter, C., et al. (2017). Emergent constraints on projections of declining primary production in the tropical oceans, 7(April), 4–9. https://doi.org/10.1038/NCLIMATE3265

Fu, W., Randerson, J., & Moore, J. K. (2015). Climate change impacts on net primary production (NPP) and export production (EP) regulated by increasing stratification and phytoplankton community structure in CMIP5 models. Biogeosciences Discussions, 12(15), 12851–12897. https://doi.org/10.5194/bgd-12-12851-2015

---

## Author Response (AR1)

**Dear Editor, Dear Professor Kravitz,**

We thank you for the opportunity to submit our revised paper to be published in ESD. We hope that with the modifications detailed below it can now be accepted for publication. Overall, based on the reviewers' comments and on your suggestions, we believe that in this new version there is an improved presentation of our results.

We reply below your comments and suggestions and indicate the new parts of the paper based on your suggestions. Replies to the reviewers' suggestions have already been uploaded, but we include them for reference.

*Editor Decision: Reconsider after major revisions (27 May 2020) by Ben Kravitz*
*Comments to the Author:*
*The reviewers have recognized that this manuscript may have some value as an idealized framework that provides context for more detailed applications. There is certainly a place for this sort of mathematical formalism. Reviewer 2 rightly points out that such formalisms need to demonstrate clear application to the real world. One example could include verifying that the model can, to some degree, reproduce real world data. Another example could include ensuring that, although the model may be missing some processes, it still contains the right ones - phrased in the converse, one needs to be certain that increasing the complexity of the model will not totally change the answers one gets.*

*The authors need to instill confidence in readers that their model is useful. I believe this could be the case, but those points are not as clearly communicated as they need to be. I encourage the authors to focus on this when revising the manuscript.*

In our paper we present a conceptual model. Its relative simplicity allows us to treat it mathematically and to carry out a detailed numerical investigations. Our conclusions are robust in a mathematical sense, meaning that small changes in our model (inclusion of noise, for example) will not alter our main findings. As long as the addition of other interactions only provide a small perturbation, our conclusions remain valid.

There is no guarantee, however, that adding more complex and extended interactions, degrees of freedom to our model will not lead to different trends. This is precisely the point of our paper: we separate the most important variables, feedbacks and interactions to reveal their effects on global trends. We do not wish to conclude that the addition of other interactions cannot lead to different trends. However, the results obtained from our model are confirmed by comparing them with the outcome of more complex models, we see in our model the same trends. In fact, the new citations included after the suggestions of reviewer 2 are probably the best confirmation that our very conceptual model can give a simple account for the trends seen in more realistic models. The difference is that the reason of these trends is transparent in our model, whereas its origin may remain hidden among the multitude of variables and interactions found in more complex models.

To further strengthen this line of thought, the following changes have been made to the text of the paper.

We added the following new text to the Introduction:

> In spite of the current trend to include biogeochemistry in climate models (see e.g. Schlunegger et al, 2019), a basic understanding of such processes is still limited. It is still under debate whether net primary production is increasing or decreasing in coupled carbon–climate models as a consequence of warming induces production increase and stronger nutrient limitations induced by increased stratification (Laufkötter et al, 2015). The situation appears to be similar to the understanding of thermal or fluid dynamical concepts decades ago. The study of e.g. the energy balance Ghil (1976) or of the thermohalin ciculation Stommel (1961) started with elementary conceptual models which later evolved into more complex ones, and are by now decisive components of cutting-edge climate models. We therefore propose here to study a conceptual atmosphere–plankton model where emphasis is on a proper choice of couplings (feedbacks).

We added the following new text into the Conclusions:

> As far as we know, our work is the first step in the direction of studying the feedbacks between the atmosphere and the biosphere by a simple conceptual model. Our conclusions are robust in a mathematical sense, meaning that small changes in our model (inclusion of noise, for example) will not alter our main findings since snapshot attractors are robust. As long as the

addition of other interactions only provide a small perturbation, our conclusions remain valid. In general, it is an open question in complex nonlinear systems whether neglected couplings to other subsystems and other simplifications could cause qualitative change in the dynamical behavior of a model. However, we see two important reasons why we believe our model goes in the right direction. First, the trends we find in our model are in accordance with the trends observed in the majority of complex models as mentioned above. Second, we believe that in our model the origin of trends are more transparent than in more complex models where this origin can be hidden among the multitude of variables, feedbacks and interations. Our model is a conceptual model, and as such, both the biological and climate models are highly simplified. However, one can consider it as a starting module of an extendable model system. On the one hand, more trophic levels and inorganic resources can be easily added to the biological side of our model, on the other hand, simple ocean circulation models can extend the climate side of our model in order to make a first step to build more complex coupled models (Daron and Stainforth, 2013). We think that mutual interactions and iterations between conceptual models and detailed Earth System Models (ESM) help to reveal the distinction between relevant and less relevant mechanisms and feedbacks behind climate change. We expect deeper insight into these feedbacks by studying conceptual and ESMs parallelly in the future.

We also added to the Conclusions the following new text:

One of our main results is that an increase in the global temperature reduces mixing intensity, which is the leading factor in decreasing the total biomass of primary producers. Interestingly, this result is in concordance with numerous studies applying Earth System Models with vastly more detailed plankton models (Bopp et al, 2013; Fu et al, 2016; Kwiatkowski et al, 2019), although other works report different observations (Laufkötter et al, 2015; Flombaum et al, 2020).

We thank you again for the comments and suggestions. The new and changed parts of the manuscript are typeset in **boldface**. We hope the presentation of our work is now acceptable to be published in ESD.

**Reply to reviewer 1**

We thank the reviewer for a thorough reading of our paper and for the useful suggestions. Below we reply them and indicate the changes made to the manuscript.

*This work presents an analysis of the feedbacks between atmosphere and ocean life with simplified mathematical models and detailed mathematical analysis. The questions and science considered in the paper are of broad relevance to researchers across many slices of the life sciences. Overall, the study is very good and offers broadly applicable insights relevant to the Earth Sciences. However, the paper does not sufficiently put the Earth Science relevant findings and broader implications front and center for ESD and its audience. The findings are there, but the paper (and especially the introduction and conclusion) would benefit from expansion in this direction. Overall, I suggest re-arranging the material to highlight broader relevance.*

We thank the reviewer for considering the study "very good" which "offers broadly applicable insights relevant to the Earth Sciences". We are grateful for his/her suggestions (below) that gives us the opportunity to highlight the broader relevance of our findings, as detailed below.

*Major Comments:*

*1. As mentioned, a greater focus on the broader Earth science issues and relevance is needed for an ESD paper. This can likely be accomplished through changes to the introduction and discussion/conclusions. Questions I wish the paper had addressed are along the lines of: what do these findings mean for more complex, process-based Earth System Models? I wanted more than Lines 355-356, and I think more could be said.*

In the revised form of the Introduction, we express our view that the understanding of the interplay between biogeochemistry and climate is still limited, and the situation of this problem is similar to the state climate science faced decades ago. This requires the use of a hierarchy of conceptual models increasing in details to shed light on the importance of various processes. Our simple conceptual model is an attempt to make the first step in this direction by coupling biogeochemistry and climate to identify the relative importance of some basic feedback mechanisms. In the Conclusions we also added that this model can be developed into a sequence of gradually more complex ones.

We added the following new text into the Introduction:

> In spite of the current trend to include biogeochemistry in climate models (see e.g. Schlunegger et al, 2019), a basic understanding of such processes is still limited. It is still under debate whether net primary production is increasing or decreasing in coupled carbon–climate models as a consequence of warming induces production increase and stronger nutrient limitations induced by increased stratification (Laufkötter et al, 2015). The situation appears to be similar to the understanding of thermal or fluid dynamical concepts decades ago. The study of e.g. the energy balance Ghil (1976) or of the thermohalin ciculation Stommel (1961) started with elementary conceptual models which later evolved into more complex ones, and are by now decisive components of cutting-edge climate models. We therefore propose here to study a conceptual atmosphere–plankton model where emphasis is on a proper choice of couplings (feedbacks).

We added the following new text into Conclusions:

> As far as we know, our work is the first step in the direction of studying the feedbacks between the atmosphere and the biosphere by a simple conceptual model. As such, both the biological and climate models are highly simplified. However, one can consider it as a starting modul of an extendable model system. On the one hand, trophical levels and inorganic resources can be easily added to the biological side of our model, on the other hand, simple ocean circulation models can extend the climate side of our model in order to make a first step to build more complex coupled models (Daron and Stainforth, 2013). We think that mutual interactions and iterations between conceptual models and detailed Earth System Models (ESM) help to reveal the distinction between relevant and less relevant mechanisms and feedbacks behind climate change. We expect deeper insight into these feedbacks by studying conceptual and ESMs parallelly in the future.

*1a. Section 3 is a good example of how the paper is heavily focused on the details of the math. There's good scientific insight there: Lines 179-181 "The relation indicates that in the case of a positive*

*enrichment parameter the phytoplankton dynamics weakens the climate change, weakens the trend from $D_0$ to $D$ in the temperature contrast, as expected. Quite surprisingly, however, the effect is rather weak since $\alpha\beta$ is quadratically small." Is there a way to make that point up front in this section, with fewer references to equations, and to move even more of the equations to the SI? Adjustments along these lines throughout would be beneficial to appeal to a broader audience of researchers.*

We relegated the derivation of our formulae into the SI, and only kept those mathematical results in the main text that are explicitly used to reach the conclusions. We added a short paragraph about what a naive expectation suggests without any mathematical treatment, then reach the conclusion by analysing the results of the detailed calculation (obtained in the SI).

We added the following new text to Sec. 3:

> Naively, one expects that an increased $CO_2$ level (smaller $F$ in (1)) leads to a higher carrying capacity and concentration of the plankton, and a slower decrease of the temperature contrast, i.e., $S(D)$ should increase (decrease) with the enrichment parameter. However, only by calculating the precise dependence can reveal whether these trends are important or hardly discernible.

*2. One easy change would be to include a table of variable and parameter notations, the quantities each notation represents, and any assumed values or boundaries imposed on the variables/parameters (such as alpha). This could be included in the SI, but is important to include, given the number of variables, parameters, and values being considered.*

We thank the reviewer for the suggestion. The table has been added as Supplementary Material III.

*3. Similarly, any kind of figure/model schematic illustrating the setup and feedbacks (and their notations where possible) would be beneficial in the main text Section 2.*

A schematic drawing, illustrating the main feedbacks used in the paper, has been added to Section 2 as a new Fig. 1. We think that this drawing indeed helps the reader by making the set of feedbacks used in the paper easier to overview. We also attach this new figure to the reply.

*Specific comments:*

*1. Lines 42-57: some of this text would be better suited in a methods section than the introduction.*

The first part of the mentioned lines provides a general qualitative introduction to the ensemble method, heavily used in our approach. The next part, describing the concept of snapshot attractors, is indeed too strongly mathematics oriented, and we hence moved it to the beginning of Section 5.

The moved sentences:

> The mathematical concepts underlying the ensemble view are snapshot (Romeiras et al, 1990) or pullback (Ghil et al, 2008) attractors. One might consider the ensemble of all permitted climate realizations over all times as the pullback attractor of the problem, and the set of the permitted states of the climate at a given time instant as the snapshot attractor belonging to that time instant (their union over all time instants is the pullback attractor). Both views express that the climate system possesses a plethora of possibilities. In the terminology of climate science, climate has a strong internal variability (e.g.Stocker et al (2013)). The concept of snapshot or pullback attractors is nothing but a reformulation of this fact in dynamical terms.

> In numerical simulations, we consider the members of an ensemble simulation to describe parallel climate realizations only after the initial conditions are "forgotten", transient dynamics disappears. Due to dissipation, this time is typically short compared to the time span of interest. Such an ensemble approach was shown to be the only method providing reliable statistical predictions in systems with underlying nonpredictable dynamics (since in this class the traditional approach based on single time series is known to provide seriously biased results). A number of papers illustrate these statements within the physics literature (see. e.g. (Romeiras et al, 1990; Lai , 1999; Serquina et al, 2008)), as well as in low order climate models (Chekroun et al, 2011; Bódai et al, 2011; Bódai and Tél, 2012; Bódai et al, 2013; Drótos et al, 2015), in general circulation models (Haszpra and Herein, 2019; Kaszás et al, 2019; Pierini et al, 2018, 2016; Drótos et al, 2017; Herein et al, 2017; Bódai et al, 2020; Haszpra et al, 2020; Haszpra and Herein and Bódai, 2020) and also in experimental situations (Vincze, 2016; Vincze et al, 2017).

*2. Line 60: What is the relevance to this work that the ensemble approach has been used in other adjacent but distinct studies that presumably consider different models of different variables?*

The ensemble method turns out to be the only reliable method in processes taking part in the presence of climate change. The traditional approach based on a single time evolution is not representative, and might lead to biased conclusions. This we emphasize now in the Introduction.

The adjusted text in the Introduction:

> An appropriate treatment of even elementary models describing climate change is not obvious since basic parameters change with time and, therefore, traditional long-time averages cannot be used to define (in the sense of any statistical quantifiers) a state of the climate. An emerging new view, already embraced by Drótos et al (2015), follows a different route to obtain information on instantaneous statistical quantifiers (e.g. expected, average properties) of the climate. Since our information on the actual state of the climate is incomplete, one imagines an ensemble of parallel Earth systems carrying parallel climate realizations subjected to the same set of physical laws, boundary conditions and external forcing, but with different initial conditions. Then the chaotic or turbulence-like properties of the climate dynamics allows for distinct climate realizations (for a review see Tél et al, 2019). These realizations, however, cannot be arbitrary since only those are permitted that are compatible with physical laws and the given forcing. The ensemble of realizations defines a probability distribution of all the relevant variables at any instant of time from which one can obtain expected, ensemble average properties of the climate (for more details, and mathematical aspects, see Sec. 5). It is therefore natural to use the ensemble view in our conceptual biogeochemistry model, too. The ensemble approach in it corresponds to generating parallel atmosphere–phytoplankton realizations from different initial conditions.

*3. Figure 8 : Could you add a colorbar rather than (or in addition to) writing it out in the caption?*

We added the colorbar to Figure 8 (now Fig. 9, due to the addition of the new schematic drawing, Fig. 1, in Section 2).

*4. Section 4: overall I like this section very much but please make explicit mention that angled brackets always correspond to ensemble average, for every variable, early in the section.*

We have made this explicit at the beginning of Section 4.

The new text added to Section 4:

> Here and in what follows angled brackets $<>$ will always denote averages taken with respect to our ensemble at a given time instant, $t$.

*5. Please consider making code and possibly some archive of the ensembles you run available to support open-access, reproducible science.*

We upload the code as supplementary information upon acceptance of the paper.

We thank again the reviewer for the insightful comments, and we hope that with the indicated changes the paper can be accepted for publication in ESD.

**Reply to reviewer 2**

We thank the reviewer for a thorough reading of our paper and for the constructive comments and suggestions that, we believe, improved the presentation of our results. We reply all comments and indicate the changes made in the manuscript.

*This manuscript describes a conceptual modelling study of how phytoplankton respond to changes in temperature, atmospheric CO2, and ocean mixing. In this study the authors attempt to use a relatively simple model that contains many assumptions to simulate phytoplankton growth as a function of changes in anthropogenic forcing. While I am not opposed to simple conceptual models that focus on key processes or dynamics, I feel that the conceptual model presented here makes many assumptions that are not well justified.*

We thank the reviewer that he is also in favor of conceptual models like the one we present here. We agree that our model only grasps the most important processes only in the simplest possible way, and neglects many effects that need to be addressed in an extended model. We believe, however, that the processes we include in the model are well justified. To better support this statement, we added a new schematic drawing (new Fig. 1, also attached to the reply) to illustrate the processes and feedbacks we include in our model. We believe that it is possible to extend our model by sequentially adding more and more details into it, as we now describe this in the Conclusions. We believe that applying this approach allows us to identify how individual components contribute to the overall behavior of complex models. We also explain this more explicitly in the Introduction.

To give more explanation why we think our model is well justified as a conceptual model, we added the following new text into the Introduction:

> In spite of the current trend to include biogeochemistry in climate models (see e.g. Schlunegger et al, 2019), a basic understanding of such processes is still limited. ... The situation appears to be similar to the understanding of thermal or fluid dynamical concepts decades ago. The study of e.g. the energy balance Ghil (1976) or of the thermohalin ciculation Stommel (1961) started with elementary conceptual models which later evolved into more complex ones, and are by now decisive components of cutting-edge climate models. We therefore propose here to study a conceptual atmosphere–plankton model where emphasis is on a proper choice of couplings (feedbacks).

We also added the following new text into the Conclusions:

> As far as we know, our work is the first step in the direction of studying the feedbacks between the atmosphere and the biosphere by a simple conceptual model. As such, both the biological and climate models are highly simplified. However, one can consider it as a starting modul of an extendable model system.

And a bit later we emphasize:

> We think that mutual interactions and iterations between conceptual models and detailed Earth System Models (ESM) help to reveal the disctinction between relevant and less relevant mechanisms and feedbacks behind climate change. We expect deeper insight into these feedbacks by studying conceptual and ESMs parallelly in the future.

*In addition, many relevant dynamics seem to be left out, e.g., other bottom-up controls on phytoplankton growth such as light and nutrients like N, P, and Fe, as well as top-down controls like grazing and mortality. While, it may be possible that global phytoplankton-climate dynamics and responses to forcing can be explained without such factors (although I doubt it), in order for me to have confidence in their model I would need to see better evidence that justifies the model simplifications and model parameterizations. Many other conceptual studies of phytoplankton use laboratory studies or observations to justify their model structure and parameterizations, surely this can be done here as well.*

We agree with the reviewer that our treatment of phytoplankton neglects many important factors, this is why we call it a "conceptual" model already in the title of the paper. We point out that our choice for the ecological component in the form of a very simple logistic equation is motivated by the similarly simple representation of the full atmosphere by merely three ordinary differential equations of the Lorenz'84 model. We feel that the coupling of this simple atmospheric model to a detailed ecological

model would be inconsistent. By taking such a simple set-up, we intend to strengthen the conclusions of more complex models by obtaining exact results in a simple transparent conceptual model. We prefer to work with a well understood, but at the same time paradigmatic description which enables one to explore the mechanism of basic feedbacks. Thus, for example, our approach makes possible to study the feedback of the primary producer on the temperature contrast which, in turn, drives the atmospheric dynamics. This might remain hidden in the complexity of current GCMs.

In the new closing section of the Conclusions we formulate that the present approach is considered to be a starting project on which a full hierarchy of models can be built. The next step in this hierarchy can be a still conceptual, combined atmosphere-ocean model to which a simple ecology model with more trophic levels and inorganic resources can be coupled.

These points are now added to the text in the Conclusions:

> On the one hand, more trophical levels and inorganic resources can be easily added to the biological side of our model, on the other hand, simple ocean circulation models can extend the climate side of our model in order to make a first step to build more complex coupled models (Daron and Stainforth, 2013).

> *In addition, and perhaps more importantly, for me to have confidence in the model there needs to be some validation, i.e., comparisons to actual data. I know that the goal is to simulate climate change and obviously one cannot validate future projections. However, it should still be possible to come up with a clever way to validate the model (or key underlying equations) using observations. There is also no real attempt to contrast the results of this study with other phytoplankton focused climate studies that have used ocean-only or Earth system models (e.g., see citations listed below). Without satisfactory justification and validation of the model and the results this is just a mathematical exercise that while interesting, leaves the reader wondering if it has meaning. Therefore, I must recommend that the manuscript be rejected.*

We thank the reviewer for the citations, we found them immensely useful to put our work into context, and we added all of them to the list of references. Papers like these give the most important motivation for conceptual models like the one we present in our paper. These papers shed light on the fact that it is still under debate whether net primary production is increasing or decreasing in coupled carbon–climate models as a consequence of warming induces production increase and stronger nutrient limitations induced by increased stratification. We also compare now our findings with the conclusions of these papers.

To sum this up, we added the following new text to the Introduction:

> In spite of the current trend to include biogeochemistry in climate models (see e.g. Schlunegger et al, 2019), a basic understanding of such processes is still limited. It is still under debate whether net primary production is increasing or decreasing in coupled carbon–climate models as a consequence of warming induces production increase and stronger nutrient limitations induced by increased stratification (Laufkötter et al, 2015).

And:

> The direct effect of increased CO2 concentration on phytoplankton dynamics can be stimulating or inhibiting, we study both scenarios.

We added to the Conclusions the following new text:

> One of our main results is that an increase in the global temperature reduces mixing intensity, which is the leading factor in decreasing the total biomass of primary producers. Interestingly, this result is in concordance with numerous studies applying Earth System Models with vastly more detailed plankton models (Bopp et al, 2013; Fu et al, 2016; Kwiatkowski et al, 2019), although other works report different observations (Laufkötter et al, 2015; Flombaum et al, 2020).

We thank the reviewer again for his insightful comments, and especially for the very useful citations that support our conclusions that global warming is expected to reduce the total biomass of primary producers. We hope that with the changes we made to the text, our paper can be accepted for publication in ESD.

[revised manuscript text omitted]